# Preleukemic Fusion Genes Induced via Ionizing Radiation

**DOI:** 10.3390/ijms24076580

**Published:** 2023-04-01

**Authors:** Pavol Kosik, Milan Skorvaga, Igor Belyaev

**Affiliations:** Department of Radiobiology, Cancer Research Institute, Biomedical Research Center, Slovak Academy of Sciences, 845 05 Bratislava, Slovakia

**Keywords:** ionizing radiation, DNA damage, leukemia, preleukemic fusion genes

## Abstract

Although the prevalence of leukemia is increasing, the agents responsible for this increase are not definitely known. While ionizing radiation (IR) was classified as a group one carcinogen by the IARC, the IR-induced cancers, including leukemia, are indistinguishable from those that are caused by other factors, so the risk estimation relies on epidemiological data. Several epidemiological studies on atomic bomb survivors and persons undergoing IR exposure during medical investigations or radiotherapy showed an association between radiation and leukemia. IR is also known to induce chromosomal translocations. Specific chromosomal translocations resulting in preleukemic fusion genes (PFGs) are generally accepted to be the first hit in the onset of many leukemias. Several studies indicated that incidence of PFGs in healthy newborns is up to 100-times higher than childhood leukemia with the same chromosomal aberrations. Because of this fact, it has been suggested that PFGs are not able to induce leukemia alone, but secondary mutations are necessary. PFGs also have to occur in specific cell populations of hematopoetic stem cells with higher leukemogenic potential. In this review, we describe the connection between IR, PFGs, and cancer, focusing on recurrent PFGs where an association with IR has been established.

## 1. Ionizing Radiation and DNA Damage

DNA damage is the most important consequence of exposure of human tissues to ionizing radiation (IR). The damage of DNA induced by IR can be: (i) direct, resulting in the DNA strand break caused by ionization and (ii) indirect, e.g., via radiolysis of H_2_O molecules generating highly reactive species including hydroxyl radicals, which can damage DNA and other biomolecules [1,2]. IR results in a wide variety of DNA molecular damage, such as base damage, single or double-strand breaks (DSBs), and DNA-protein cross-links, some of which may be clustered to form complex lesions. One Gray (Gy) of IR induces approximately 2000 base modifications [3], 1000 DNA single-strand breaks [4], and 35 DSBs per cell [5]. It is widely accepted that DSBs are the most serious lesions capable of causing cell death or cell transformation [6,7]. In human cells, DSBs are repaired by non-homologous end-joining (NHEJ) or homologous recombination (HR) [8]. Misrepair of the DSBs can result in the formation of chromosomal aberrations, including fusion genes, that in turn can lead to the development of various diseases and cancers [9,10]. IR also produces a wide range of effects with potential implications for carcinogenesis, e.g., modulating protein phosphorylation [11], gene expression [12], altering epigenome by producing DNA methylation [13,14], histone methylation [15], and chromatin modification [16,17].

## 2. Mechanisms of Fusion Genes Formation and Its Prevalence

Fusion genes are formed via somatic chromosomal rearrangements. There are four basic forms of chromosomal rearrangements (i) translocations, (ii) deletions, (iii) insertions, and (iv) inversions [18]. In the case of translocations, one or two chromosomes are involved, generating non-reciprocal or reciprocal fusion products. For other rearrangements only one chromosome is usually involved.

In general, chromosomal translocation requires formation of DSBs. Several mechanisms have been proposed for recurrent balanced chromosomal translocations observed in leukemic patients. They include: (i) illegitimate V(D)J or switch recombination; (ii) homologous recombination mediated by repetitive sequences, e.g., Alu and LINE elements; (iii) DNA topoisomerase II subunit exchange; and (iv) error-prone NHEJ [19,20]. Several DNA sequences (DNA hotspots) have also been suggested to elevate the occurrence of DSBs. For leukemia, these recombination hotspots are formed by the purine/pyrimidine repeat regions [21,22,23], scaffold/matrix attachment regions, and preferential DNA TOPO II cleavage sites [24,25].

Additionally, two chromosomes producing translocations must come in contact with one another. Therefore, the respective DNA loci should be in close spatial proximity to each other during the interphase. Several studies have supported this hypothesis [26,27,28,29]. For example: (i) 3D fluorescent in situ hybridization analysis has shown that mixed-lineage leukemia (MLL) and its translocation partners, eleven-to-nineteen leukemia proteins (ENL) and AML-fused genes from chromosome 4 (AF4), are adjacently located in interphase nuclei [26]; (ii) BCR and ABL loci are close to one another in CD34^+^ bone marrow cell [27]; and (iii) RET and PTC1 loci involved in RET–PTC1 fusion in papillary thyroid cancer were found to be closer to one another in thyroid cells than in lymphocytes [28,30,31].

In addition, fusion transcripts (FTs) can also arise via abnormal transcription yielding transcript chimeras, although relevant chromosomal genes are intact. These FTs can be induced by two mechanisms: (i) trans/cis-splicing [32,33] and (ii) read-through [34]. There are rare reports of FTs generated by non-structural genomic rearrangement mechanisms, e.g., transcription read-through and intergenic splicing, but the functions of these FTs are poorly understood.

Jividen and Li [35] found JAZF1-SUZ12 fusions as canonical FTs (expressed from chromosomal translocation t(7;17) in human cancer cells and as non-canonical chimeras in non-malignant human samples, with the resultant FTs being identical. The authors suggested that the non-malignant chimeras may be related to the stress response in cells exposed to hypoxia. This hypothesis was supported by data showing stress-induced apoptosis in HEK293 cells with forced expression of this fusion. PAX3-FOX01 FTs represent another example [36]. FTs encoded by chromosomal translocation t(2;13) are a driving force of alveolar rhabdomyosarcoma, while the identical fusion chimeras are found in muscle biopsies in aborted fetuses, i.e., they have a developmental role in normal cells [33,36]. It means that identical FTs can be formed by trans-splicing (non-malignant) and chromosomal rearrangement (neoplasia). Suggested by the authors, Model 1 (“true-true, unrelated”) assumed that some factors facilitate chromosomal translocations and trans-splicing, while Model 2 (“*trans*-splicing mediation”) proposed that trans-spliced RNA facilitates chromosomal translocation. Zhang et al. [37] studied chimeric SLC45A3-ELK4 RNA generated in the absence of corresponding chromosomal rearrangement and revealed that it is formed via the cis-splicing of adjacent genes, i.e., via read-through. The levels of chimeric RNA correlated with disease progression, i.e., the highest levels were detected in prostate cancer metastases. The mechanism of its overexpression in metastases remains unknown. This cis-SAGe, i.e., cis-splicing of adjacent genes, may represent a novel epigenetic way to drive cancer.

With the development of new laboratory techniques, the number of known fusion genes has rapidly increased. In the last two decades, next generation sequencing provided a novel approach for the detection of fusion genes at the DNA or RNA levels [38,39,40]. Until now (20 August 2022), the Mitelman database of chromosome aberrations and gene fusions in cancer https://mitelmandatabase.isb-cgc.org (accessed on 20 August 2022) contains more than 70,718 cases, over 32,962 gene fusions, and more than 14,016 genes [41].

It has been demonstrated that leukemias (and solid tumors) harboring chromosomal translocations may also contain aberrant fusion-circular RNAs (f-circRNAs) [42,43]. Based on hypothesis, the formation of f-circRNA made by the fusion of two translocated genes is allowed by the juxtapositioning of complementary sequences in introns upstream and downstream of the breakpoint region of the translocation [44,45]. It has been shown that bone marrow-derived leukemic cells of acute promyelotic leukemia (APL) patients contained f-circ PML-RARA (f-circPR) with a backsplice junction between PML exon 5 and RARA exon 6. However, three out of four patients expressed an additional, alternative f-circPR (backsplice junction between exon 4 in both PML and RARA gene) [44]. These data suggest that a fusion gene resulting from chromosomal translocation can give rise to one or more f-circRNAs. Similarly, expression (generation) of multiple circRNAs was also shown from MLL-AF9 fusion genes (f-circM9) [44]. “Multiple hits in one” scenario assumes that concomitant expression in cancer cells of linear fusion mRNA, fusion proteins, and f-circRNAs could be critical for cancer onset and progression [46]. The expression of f-circRNAs in cancer cells is essential for their maintenance and confers resistance to therapy, for e.g., f-circM9 expression was shown to protect cancer cells from drug-induced apoptosis [47].

## 3. IR Induced Leukemia

Leukemia is a clonal disease which occurs after genetic transformation of the hematopoietic stem, progenitor, and precursor cells [48,49,50]. It is the most frequent cancer in children. According to a generally accepted hypothesis, childhood leukemia results from at least two consecutive events [51]. The first hit is usually prenatal chromosomal translocation that leads to preleukemic fusion genes (PFGs) (Figure 1), and the second hit usually represented by point mutations, deletions, or insertions often occurs postnatally [52,53]. The question is whether exposure of females during pregnancy, e.g., during radiological examinations, can contribute to formation of PFGs in their newborns. Childhood ALL and AML are distinct from those occurring in adulthood with respect to molecular (e.g., cytogenetic) and other features, and in adults there may be no prenatal mutational component [54,55]. Moreover, it is also possible that the “preleukemic” fusion event itself is actually the “second hit”, preceded by another, perhaps the single nucleotide mutation in another, unidentified oncogene or tumor suppressor gene. In this sense, the “first hit” and the “second hit” may be interchangeable in terms of sequentiality.

Leukemia, similarly to other cancers, has been associated with IR. This information comes mainly from large epidemiological studies, especially the Life Span Studies (LSS) on the atomic-bomb survivors [57,58]. The linear and linear quadratic models are the commonly accepted models for radiation response. The linear quadratic model (Y = C + αD + βD^2^) is based on the hypothesis of dual radiation action. Y is the yield, D is the dose, C is the control (background frequency), α is the linear coefficient, and β is the dose-squared coefficient. The linear component is produced by a single act of radiation and the quadratic component is the result of two independent acts of ionising radiation. In a range of very low doses the contribution of quadratic components is negligible, so the dose response can be approximated with linear components. For higher doses the quadratic component is increased. While a linear dose response pattern was described for most solid cancers [59,60], the leukemia incidence and mortality data better fitted to the linear quadratic dose response [61,62,63]. Analysis of the type of genetic alterations produced by radiation can provide clues to follow the etiology of cancer in exposed populations. At low doses, the principal radiation-induced mutations are point mutations. With an increasing dose, deletions became the predominant mutations [64]. The dose response for induction of point mutations is linear, while that for deletions fits a linear quadratic response [64,65]. Chromosomal aberrations as dicentric chromosomes have a linear quadratic dose response [66]. In case of chromosomal translocations, the data are not consistent and suggest a linear [67,68] or linear quadratic dose response [69,70]. In view of incidence and mortality data for IR induced leukemia, a linear quadratic dose response of chromosomal translocations might better indicate their involvement in IR-induced leukemogenesis. On the basis of the Japanese bomb survivor data, the lifetime risk for developing leukemia after acute whole-body exposure to 1 Gy is estimated to be 0.85%, i.e., 85 per 10,000 persons [71,72]. Depending on age and time after exposure, radiation affects induction of leukemia subtypes in different ways. In the study of Preston et al., the excess relative risk (ERR) estimates for the leukemia subtypes at 1 Sv were 9.1, 3.3, and 6.2 for ALL, AML, and CML, respectively [73]. The LSS studies of atomic bomb survivors provided primordial association between IR exposure to a single moderate dose and cancer using a large population of all ages. However, their power to assess the cancer risk from a very low and repeated dose exposure is very limited.

Nowadays, more than a half of the current exposures to low doses of ionizing radiation come from radiological procedures [74,75]. Associations between cancer and radiological examinations were accessed by a number of epidemiological studies [76,77]. Although these epidemiological studies have some limitations and do not match each other in all aspects, the obtained data suggest that even very low IR doses may increase the risk of developing leukemia [78]. In a study of Pearce et al. [79], 0.036 ERR/mGy (*p* = 0.0097) of leukemia was detected after irradiation from CT scans. In the same study, the relative risk of leukemia for patients who received a cumulative dose of at least 30 mGy compared to patients who received 5 mGy was increased to 3.18 (95% CI 1.46–6.94).

A significant excess risk of leukemia was also associated with natural environmental radiation exposure [80,81]. In the UK, the average exposure to natural sources of ionizing radiations including cosmic rays represents 2.6 mSv/yr, suggesting that approximately 6% of all leukemia might be attributable to the natural radiation exposures [71].

Another cause of leukemia induced via IR may be radiation therapy (so-called secondary or therapy-related leukemia) [82,83]. According to the study by Wright et al. [84], from the cohort (199,268 individuals) that included 66,896 (33.6%) patients who received pelvic radiotherapy, 144 (0.215%) patients developed secondary leukemia. The most patients developed AML (56%), less developed CML (20%), and the remaining cases represented ALL. In the Cox proportional hazards model, the risk of post-treatment leukemia was increased by 72% (HR, 1.72; (95% CI, 1.37–2.15)) in the patients who received radiotherapy compared with those who were not exposed. Of note, the frequency of leukemia may be biased by shortened life expectancy due to the first (unrelated) malignancy in these cohorts, given the relatively long latency of the iatrogenic leukemia.

In contrast to the extensive data that consider all the causalities of leukemia worldwide, in this review we focus on the IR-induced PFGs. We searched the PubMed database to find all published experimental studies estimating IR induction of PFGs both in vitro and in vivo. We used various combinations of search terms such as “fusion genes”, “preleukemic fusion genes”, “ionizing radiation”, “induction”, and “leukemia”. The time period of the searched articles—with English language restriction—ranged from 1945 to September 2022.

## 4. PFGs Induced by IR

### 4.1. Induction of BCR-ABL1 Gene Fusions

BCR-ABL1 is generated from reciprocal translocation between chromosome 9 and 22. This PFG is present in 95% of CML patients and less often in patients with AML and ALL. According to the breakpoints in the three main breakpoint cluster regions of the BCR gene, namely major (M-bcr), minor (m-bcr), and micro (mu-bcr), at least eight different fusion transcripts, varying in size, can be produced. These three regions are usually associated with p210, p190, and p230 fusion proteins, respectively [85].

There are few in vitro and in vivo experimental studies that show that IR induces BCR-ABL1. For example, Ito et al. [86] used the HL-60 cell line derived from myeloid leukemia and irradiated actively growing cells in Petri dishes with X-rays (100 Gy). After 48-h incubation in a growth medium, the total RNA was isolated and 10^8^ affected cells were analyzed in a total of 41 nested reverse-transcription PCRs (R-T PCRs). They found 5^+^/41 reactions and identified two types of fusions via sequencing: (i) CML-specific BCR-ABL1 p210: b2a2, b3a2 fusions and (ii) atypical fusions which were probably non-functional.

Spencer et al. [87] used karyotypically normal lymphoblastoid cell lines (LCL) derived from four leukemic patients (two AML, two CML) and two healthy individuals. They irradiated cells with low energy transfer (LET) γ-radiation (50, 100 Gy). After incubation in growth media for 18–28 h, the total RNA was isolated and reverse transcribed into cDNA that served as a template for nested PCR. 

The data show that all LCLs contained a background level (at 0 Gy) of BCR-ABL1 p210. There was no increase in BCR-ABL1 in LCL derived from the healthy individuals, while statistically significant induction of BCR-ABL1 was found in LCL derived from the AML/CML patients. Referring to other studies [88,89], this study assumes that the detection of residual BCR-ABL positivity within CML-derived LCL was unlikely and did not produce false-positive PCR results. However, contamination still cannot be excluded.

Mizuno et al. [90] carried out in vivo experiments using human fetal thyroid tissue transplanted into a SCID mouse and allowed to grow for 17 months when it was locally irradiated with 50 Gy of X-rays. The affected tissue was screened for the presence of PFGs; in addition to BCR-ABL1, the H4-RET fusion gene (associated with thyroid cancer) was also determined in various intervals (2 and 7 days, 2 months) after irradiation. Regarding BCR-ABL1 p210 associated with CML, fusion transcripts with typical and atypical gene rearrangements were already identified 2 days after irradiation. However, these PFGs disappeared in samples collected for analysis in later time intervals (7 days, 2 months).

Previously, BCR-ABL1 p210 fusion proteins, which lead to deregulated tyrosine kinase activity, were exclusively found only in AML/CML. The Mizuno et al. study showed that BCR-ABL may also occur in non-hematopoietic cells. These results also indicated that H4-RET and BCR-ABL1 genes, which can be induced via X-rays in cultured cells [31,86], can also be induced in vivo.

In another study, Mizuno and colleagues [91] used four cell lines: 8505C—derived from thyroid gland carcinoma, Daudi—from Burkitt lymphoma, G-401—from Wilms tumor, and HT1080—from fibrosarcoma. The cells were irradiated with X-rays with doses of 10, 50, and 100 Gy. The 10^8^ cells divided into 10 tubes were tested for each cell line and dose. The authors found a linear dose response of a BCR-ABL1 p210 FT induction, with small differences depending on the cell lines (Table 1).

Tanaka et al. [92] studied chromosomal abnormalities and *bcr*-locus rearrangements in seven CML atomic bomb survivors and compared with fourteen CML unirradiated patients. They found that all CML patients contained BCR-ABL1 p210 fusion protein. However, there was no difference in the distribution of breaks within *BCR* and *ABL* genes (b2a2 or b3a2).

In our recent studies we analyzed the induction of PFGs in mononuclear cells (MNCs) from umbilical cord blood (UCB) [93] and subpopulations of hematopoietic stem and progenitor cells (HSPCs) (Table 2) [94]. In the first study [93], the G0 non-stimulated MNCs 3 h and 24 h post-irradiation were analyzed for BCR-ABL1 p190, ETV6-RUNX1 (TEL-AML1), RUNX1-RUNX1T1 (AML1-ETO), and KMT2A-AFF1 (MLL-AF4) via real-time qPCR (RT qPCR) and validated via sequencing. Samples were analyzed in triplicate RT qPCR reactions and categorized into groups according to IR doses of 0.1, 0.5, 2, 5, 10, and 30 Gy. Neither time after irradiation nor dose was a statistically significant factor for PFG induction (*p* > 0.05). However, this study showed the low but statistically significant inducibility of BCR-ABL p190 at low doses up to 50 cGy in comparison with high doses above 50 cGy. In the second study [94], MNCs were exposed to the dose of 50 cGy, sorted to eight HSPC subpopulations using appropriate CD markers, stimulated into the cell cycle, and analyzed via RT qPCR followed by sequencing. The data did not reveal a cell population with a higher sensitivity to the formation of radiation-induced PFGs (RUNX1-RUNX1T1, KMT2A-MLLT3 (MLL-AF9), and PML-RARA), responsible for the genesis of AML (Table 2).

### 4.2. Induction of RUNX1-RUNX1T1 Gene Fusion

The RUNX1-RUNX1T1 hybrid gene is the result of fusion between chromosome 21 and 8, t(8;21)(q22;q22). The RUNX1-RUNX1T1 fusion protein acts by blocking differentiation and inducing self-renewal of hematopoietic progenitor cells [95].

Deininger et al. [96] carried out in vitro studies and observed a selective induction of PFGs following irradiation with high doses of IR. They used two hematological (myeloid) cell lines, HL-60 and KG1. Cells were irradiated with 50 and 100 Gy of γ-radiation, grown for 24 h and 48 h post-irradiation, and screened via the nested R-T PCR with a sensitivity of ~1 positive cell/10^7^ cells. In addition to RUNX1-RUNX1T1, they also screened t(9;22)(q34;q11) BCR-ABL1 and t(6;9)(p22;q34) DEK-NUP214 (DEK-CAN). They found a significant variability in PFG induction between these two studied cell lines. 

A high induction of RUNX1-RUNX1T1 (25/40), i.e., 62.5% positivity, was found but only in the KG1 cell line (*p* < 0.0001). On the other hand, there was no significant difference in the incidence of studied PFGs between the unirradiated and irradiated HL-60 cell lines as well as for BCR-ABL1 and DEC-CAN FTs. For all PFGs, both typical and non-typical joins were observed. These data suggest that IR is capable of inducing PFGs with different incidence in cell populations of different susceptibilities.

### 4.3. Induction of KMT2A (MLL)—Translocations

Chromosomal rearrangements involving the histone lysine [K]-MethylTransferase 2A (KMT2A) gene on chromosome 11q23, formerly known as the mixed-lineage leukemia (MLL) gene, are found in 10% of ALL and 2.8% of AML patients, with the highest incidence in infants [97]. The incidence of AML with MLL rearrangement (MLL_r_) was significantly higher in therapy-related AML (t-AML) than in de novo AML [98]. In total, more than 240 different MLL rearrangements have been identified so far, with 35 translocation partner genes occurring recurrently [99]. However, of them only seven MLL fusions (with AF4, AF6, AF9, AF10, ENL, ELL, EPS15) and partial tandem duplications (PTDs) encompass ~90% of all diagnosed MLLr leukemia patients [100]. It has been demonstrated that inhibitors of various crucial enzymes of DNA metabolism, such as topoisomerase II and I, DNA polymerase, and alkylating agents, as well as apoptotic stimuli, secondary DNA structures, and ionizing radiation, are able to induce MLL breakage, emphasizing the exceptionally high fragility of the MLL breakpoint cluster region [101,102,103].

Klymenko et al. [104] studied whether translocations involving the MLL gene are more frequent in radiation-associated AML when compared to spontaneous AML. Out of 61 AML patient samples with KMT2A-abnormalities revealed via FISH/R-T PCR, 27 were exposed to radiation during the Chernobyl accident and 32 were not irradiated, thus representing spontaneous AML caused by other than IR factors. The MLL translocation was found only in spontaneous AML (1/32). These data suggest no involvement of the MLL gene in radiation-induced leukemogenesis. Interestingly, in affected patients, 1/27 contained MLL duplication, supporting the assumption that loss/gain of chromosomal material could play a more important role in the leukemogenesis of AML patients exposed to radiation during the Chernobyl accident. In line with these data, our data did not reveal any induction of KMT2A-AFF1 in the IR-exposed UCB cells [93].

### 4.4. Induction of PML-RARA—Translocations

The *PML-RARA* PFG is the most critical event involved in the pathogenesis of acute promyelocytic leukemia. This derives from a cytogenetic translocation t(15;17)(q24;q21) leading to the rearrangement of PML and RARA genes [105]. PML-RARA produces a block of myeloid differentiation at the promyelocytic stage [106].

Quina et al. [107] studied the induction of PML-RARA in a lymphoid cell line. The IM9 cells were irradiated with 10 Gy and the incidence of PML-RARA transcripts was analyzed via nested PCR using two different primers. In summary, a similar incidence of PML-RARA was observed between the irradiated and control samples (*p* > 0.05). Thus, the data indicate that the PML and RARA genes are not particularly susceptible to the clastogenic effect of IR.

## 5. Probability of BCR-ABL1, RUNX1-RUNX1T1, and PML-RARA Induction

Using the same approach as described by Holmberg [108], we have estimated the relationship between radiation-induced PFGs and chronic myelogenous leukemia of atomic bomb survivors in Hiroshima and Nagasaki and a general non-irradiated population. First we estimated the frequency of balanced reciprocal translocations at a fixed radiation dose and calculated the probability that a bone marrow HSPC will contain a random radiation-induced t(9;22), t(8;21), and t(15;17) reciprocal translocation with its break points confined to the same regions as in the BCR-ABL1, RUNX1-RUNX1T1, and PML-RARA translocations, respectively. We then included the non-random mutational rate within the genome and used the calculated probability to estimate the number of individuals among the atomic bomb survivors carrying a stem cell with a radiation-induced BCR-ABL1 translocation. Finally, we compared the estimated number of individuals with the number of affected individuals with CML in Hiroshima and Nagasaki and the general non-irradiated population. To adjust for errors mentioned in the Holmberg study, newer data were incorporated for the sizes of break point cluster regions, numbers of translocation/Gy, and the number of CD34^+^ HSPCs in bone marrow.

Under the assumptions of the random distribution of IR-induced DNA strand breaks through the PFG break point cluster regions and total genome, the proportion of BCR-ABL1, RUNX1-RUNX1T1, and PML-RARA in the total yield of all translocations was estimated (A).

For BCR-ABL1:A = 4 × (200 × 10^3^ × 3 × 10^3^)/(6.4 × 10^9^)^2^ = 5.8 × 10^−11^B = 5.8 × 10^−11^ × 0.057 = 3.3 × 10^−12^C = 3.3 × 10^−12^/(20 × 10^6^ × 1400 × 10^−2^) = 1.18 × 10^−4^

For RUNX1-RUNX1T1

A = 4 × (25 × 10^3^ × 15 × 10^3^)/(6.4 × 10^9^)^2^ = 3.6 × 10^−11^B = 3.6 × 10^−11^ × 0.057 = 2.05 × 10^−12^C = 2.05 × 10^−12^/(20 × 10^6^ × 1400 × 10^−2^) = 7.3 × 10^−5^

For PML-RARA

A = 4 × (2.8 × 10^3^ × 15 × 10^3^)/(6.4 × 10^9^)^2^ = 4.1 × 10^−12^B = 4.1 × 10^−12^ × 0.057 = 2.3 × 10^−13^C = 2.3 × 10^−13^/(20 × 10^6^ × 1400 × 10^−2^) = 8.2 × 10^−6^

Along with size of the break point cluster regions for ABL1 (200 Kbp) and BCR (3 Kbp), we took into account the break point cluster regions for the RUNX1 (25 Kbp), RUNX1T1 (15 Kbp), PML (2.8 Kbp), and RARA (15 Kbp) genes and the size of the whole human genome (6.4 Gbp). Factor 4 accounts for the fact that the cell has a diploid number of chromosomes.

A linear dose response was observed between ionizing radiation doses and chromosomal translocations in lymphocytes, with 0.057 translocations yielded per whole genome cell equivalent per Gy [109]. Consequently, the probability per cell and Gy for the induction of the t(9;22), t(8;21), and t(15;17) translocations was calculated to be 3.3 × 10^−12^, 2.05 × 10^−12^, and 2.3 × 10^−13^, respectively (B).

The human bone marrow represents approx. 1.4 kg (1400 cm^3^) of the 70 kg human body [110], while the total number of nucleated cells (CD45^+^) is approx. 20 × 10^6^ mL^−1^ [111]. Of them, CD34^+^ HSPCs, which are supposed to be the cells of origin for leukemia, constitute approx. 1% (1 × 10^−2^) [112,113,114,115,116]. Accordingly, the probability for the induction of BCR-ABL1, RUNX1-RUNX1T1, and PML-RARA translocations in one of the bone marrow CD34^+^ HSPCs, respectively, was calculated to be 1.18 × 10^−4^, 7.3 × 10^−5^, and 8.2 × 10^−6^ per Gray and individual (C). Of note, the C value estimated by us for BCR-ABL1 was slightly lower in comparison to Holmberg’s data (C = 0.007) due to the updated numbers, as described above. For further estimations, we will solely consider BCR-ABL1 because almost all (95%) CML patients carry the BCR-ABL fusion gene. Taking into account the data for CML cases among the survivors of the atomic bombs at Hiroshima and Nagasaki [117], and transforming these data as suggested by Holmberg [108], a total of 9196 persons received a mean organ dose equivalent to bone marrow of 0.85 Sv. Using the number of 1.18 × 10^−4^ estimated by us for BCR-ABL1 in CD34^+^ HSPC per individual and Gy, we calculate the number of individuals (D) among the atomic bomb survivors carrying a stem cell with radiation-induced t(9;22) translocation resulting in BCR-ABL1:D: 1.18 × 10^−4^ × 0.85 × 9196 = 0.92 ≈ 1 

However, this calculation did not take into account the fact that genetic mutations are influenced by the sequence context, structure, and genomic features, what might actually lead to a ~100-fold difference in germline mutation rates across human DNA [118]. It was suggested that the PFG break point cluster regions represent the fragile parts of DNA [101]. Thus, the probability of inducing DSBs could be higher than that calculated above. Another question deals with the probability that two loci of the two PFG partners are able to rejoin. This estimation is based on the assumption of a random distribution of the radiation-induced break points. If we take into account this factor, the number of individuals (D) with radiation-induced BCR-ABL1 among the atomic bomb survivors at the highest mutation rate could be:D: 0.92 × 100 = 92 out from 9196 (i.e., ≈ 1%)

Of note, the reported number of CML cases in atomic bomb survivors in Hiroshima and Nagasaki were 18 [119], thus indicating that not all radiation-induced BCR-ABL1 are able to result in overt leukemia. Similarly, the prevalence of various PFGs, including BCR-ABL1 in the UCB of healthy newborns was about 1% [120,121], while prevalence of overt leukemia was 100 times lower [122], showing that a similar proportion of BCR-ABL1 positive individuals in general non-irradiated population and in irradiated persons (about 1%) will develop overt leukemia.

## 6. Discussion

The first radiation accident study aiming to estimate the effect of radiation on humans came from the 1960s. The presence of chromosome abnormalities was found in the peripheral lymphocytes of fishermen exposed to radiation [123,124]. To discover more about the genesis of leukemia, experimental studies using rats were conducted. From the bone marrows of irradiated rats, the clones of cells with chromosome abnormalities were obtained. Of them, a high proportion contained reciprocal translocations or inversions [125]. However, follow-up studies lasting more than 25 years did not show any direct evidence connecting radiation-induced chromosome abnormalities to the development of leukemia [126]. Later, in the 1990s, more studies considering direct IR induction of specific recurrent preleukemic fusion genes started to appear [86,87,90,91,93,94,96,107], as summarized in Table 3.

The study by Ito et al. [86] observed the induction of BCR-ABL1 PFG in the HL-60 cell line, while Deininger et al. [96] did not see any induction of BCR-ABL1 or DEC-NUP214 PFGs in the same HL-60 cell line. Unexpectedly, BCR-ABL1 genes were detected in thyroid tissue cells at 2 days after exposure, but not at 7 days or 2 months in the study by Mizuno et al. [90]. While chromosomal translocations are considered to be stable aberrations, which might persist up to one year after radiotherapy [127], this study reported their disappearance seven days or two months after irradiation. Spencer et al. [87] detected radiation-induced BCR-ABL1 in AML- and CML-derived LCL but not in LCL derived from healthy probands. RUNX1-RUNX1T1 was induced in KG1 cells but not in HL-60 in the study by Deininger et al. [96]. PML-RARA was not induced in the IM9 cell line in the study by Quina et al. [107]. In conclusion, the available studies indicate that PFG formation is essentially cell type-dependent. The higher susceptibility of selected cell lines for PFG formation is probably caused by the specific mutual position of chromosomal territories and gene proximities in different cell types [27,128]. The induction of PFGs was clearly observed at high doses ≥10 Gy and mostly at 100 Gy. Of them, BCR-ABL was induced in many cell lines [86,87,90,91], while a linear dose response (10 to 100 Gy) was found only in one study by Mizuno et al. [91]. A dose response was not observed for other PFGs. Otherwise, while many studies showed an increase in leukemia after very low doses in occupationally exposed persons or people after radiological examination, up to date, only one aforementioned in vitro study by Kosik et al. [93] showed induction of BCR-ABL1 at doses ≤0.5 Gy. However, these data are not sufficient to make any final conclusion and warrant further investigation of PFG induction after low dose exposure.

Several studies clearly showed that the frequency of PFGs in healthy newborns significantly exceeds (≥100 times) the incidence of the corresponding leukemia subtypes [120,121,129,130]. In 2005, Nakamura hypothesized that radiation-related leukemia arises only in a small number of individuals which already contain clonally expanded preleukemic cells. According to this hypothesis, the risk of leukemia development caused by radiation is not directly determined by radiation-induced PFGs, but by secondary mutations induced in preleukemic cells [131]. From that perspective, the key factor for individual risk assessment of radiation-induced leukemia is whether the individual is a carrier of clonally expanded preleukemic cells. The multi-color fluorescent in situ hybridization (mFISH) of peripheral blood lymphocytes has recently revealed a significant increase in frequencies of aberrant clones (reciprocal translocations and inversions) of in vivo irradiated patients (accidentally or therapeutically) [132,133]. Observation of clonal translocations in these patients suggests that a single stem cell of an adult can generate long-lived myeloid and lymphoid progeny. Nikitina et al. [132] has repeatedly for many years analyzed the frequency and the spectrum of persistent chromosomal aberrations in peripheral blood lymphocytes of Chernobyl Nuclear Power Plant (NPP) accident victims after acute accidental exposure to high-dose γ-radiation. This study detected four aberrant cell clones, each carrying one balanced translocation at a frequency of up to 2.5% of the analyzed cells, namely t(4;8), t(14;15), t(2;2), and t(1;15). The regions that accumulated break points were regarded as ‘hotspots,’ including 1p32–p36.1, 3p21–22, 5q31–q35, 6p21–p22, 8q11.2–q13, 10q24–q26, 12p13, 14p10–q13, and 14q24. Nakano et al. [133] also detected 96 clones composed of at least 3 cells with identical aberrations in the T lymphocytes of 513 atomic bomb survivors. Several clonal translocations have been observed, i.e., t(2;12), t(1;8), t(4;6), t(5;13), t(2;4), t(1;21), t(11;16), t(18;20), t(3;4), t(1;12), t(2;2), and t(2,14). George et al. [134] studied the occurrence of chromosomal aberrations in the blood lymphocytes of astronauts. They identified clonal cells in two of the twelve individuals after flight. In astronaut A1, a clonal translocation between chromosomes 2 and 4 were found. All these revealed clonal aberrations have not been associated with recurrent PFGs. However, the t(2;4) reciprocal translocation has been detected in patients with various types of leukemia, including ALL, AML, CLL, CML, and acute megakaryoblastic leukemia [41]. Moreover, t(1;15)(q21;q22) is rearranged in 1/3 of multiple myeloma patients and can influence prognosis [135]. These studies indicated a possible correlation with IR-induced aberrations and PFGs.

We have studied the effect of ionizing radiation on various subpopulations of UCB cells, including mononuclear cells and CD34^+^ HSPCs [94]. In general, we have not found a statistically significant induction of PFGs (Table 2). Moreover, we observed the presence of PFGs in some non-irradiated samples, but also its absence after these cells were exposed to γ-rays. This seemingly confusing result is likely due to a very low detected copy number of PFGs (1 × 10^−5^), which is close to the sensitivity threshold of the RT qPCR [121]. In these studies (Table 3), IR-induced PFGs have been analysed using either nested R-T PCR or RT qPCR, and the PCR products were used as templates for sequencing. The sensitivity of nested R-T PCR and RT qPCR has been reported in the range of 10^−6^ to 10^−7^ and 10^−4^ to 10^−5^, respectively. In most studies (Table 3) RNA was amplified separately in individual tubes containing 10^6−7^ cells. Thus, the frequencies of induced fusion genes were given as the number of tubes that showed PCR amplification per total number of tubes or reactions. Because only few tubes from the total number of tested tubes usually contained PFGs, it can be assumed that the frequency of IR-induced PFGs is very low and probably on the border of PCR sensitivity. Mizuno et al. suggested that the induction frequency for BCR-ABL1 was 1.1 × 10^−7^ in human thyroid cells after 50 Gy in one study [90] and 1 × 10^−9^, 1.4 × 10^−8^, and 4.2 × 10^−8^ at 10, 50, and 100 Gy, respectively, in four different cell lines, 8505C, Daudi, G401, and HT1080, in another study [91]. However, the crucial question to be answered is whether the induction of PFGs following exposure to IR would be increased in the HSPC subpopulations already exhibiting genome instability, e.g., evoked by oxidative stress [94,121].

## 7. Conclusions and Perspective

A number of epidemiological and experimental studies provide the key to the mechanisms of radiation-induced leukemia. In general, it is accepted that DSBs, especially their complex forms, are likely the most significant lesions induced by IR. Experimental studies show that the complex DSBs are responsible for consecutive molecular and cellular effects of IR. They are probably repaired by error-prone repair pathways, especially NHEJ, that lead to the induction of chromosomal mutations. This review found that high doses of around 50–100 Gy were mostly analyzed and at some circumstances (i.e., cell type, dose) could induce PFGs. However, the data are fragmentary and not always consistent. From all available studies, only one showed linear dose response induction for BCR-ABL1 in the dose range of 10 to 100 Gy. In addition to the absorbed dose, significant dependence on cell type was reported, likely suggesting the importance of the spatial nuclear organization of chromosome territories. A number of radiological studies suggest that leukemia might occur even after low doses of ≤0.1 Gy. Induction of PFGs by low doses was poorly investigated and further efforts are required to study low dose effects on PFG induction. The data collected in this review partially support the Nakamura hypothesis, suggesting that radiation-induced leukemia arises only in a small number of individuals which already harbor clonally expanded preleukemic cells. However, we assume that leukemia induced via ionizing radiation occurs regardless of whether first hit (PFG) or secondary hit (deletion, point mutation) or both are induced via IR, as far as assuming that both hits have to occur. Of note, our calculations, which are based on the Holmberg approach, indicated that not all radiation-induced BCR-ABL1 are able to result in overt leukemia. The same proportion of BCR-ABL1-positive individuals in the general population and in irradiated persons (about 1%) will develop overt leukemia, indicating the similar mechanism of leukemogenesis.

## Figures and Tables

**Figure 1 ijms-24-06580-f001:**
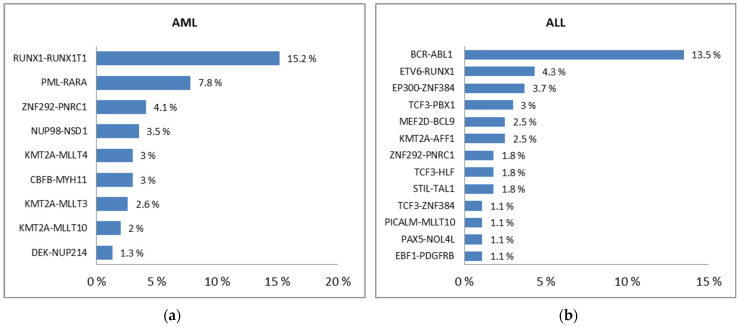
PFGs with the incidence higher than 1% in AML (**a**) and ALL (**b**) [56].

**Table 1 ijms-24-06580-t001:** Percentage of BCR-ABL^+^ samples calculated using the data reported in [91].

	0 Gy	10 Gy	50 Gy	100 Gy
BCR-ABL1 in all cell lines	0%	5%	14.2%	41.9%
8505C	0%	0%	50%	20%
Daudi	0%	0%	0%	30%
G-401	0%	10%	10%	27.3%
HT1080	0%	10%	20%	60%

**Table 2 ijms-24-06580-t002:** Induction of PFGs in different HSPC populations. Number of positive reactions combined from four UCB probands is shown (adapted from [94]).

	RUNX1-RUNX1T1	PML-RARA	KMT2A-MLLT3
Po1-C	0/6	ND	0/6
Po1-I	0/6	0/3	0/6
Po2-C	0/9	0/6	0/9
Po2-I	0/9	0/6	0/9
Po3-C	1/9	0/9	0/9
Po3-I	0/9	0/9	0/9
Po5-C	0/12	0/9	0/12
Po5-I	0/12	0/9	0/12
Po6-C	0/9	0/12	0/12
Po6-I	0/9	0/12	0/12
Po7-C	0/12	0/12	0/12
Po7-I	0/12	0/12	0/12
Po8-C	0/9	0/12	0/9
Po8-I	3/12	0/12	0/12

Abbreviations: ND, not determined; C, control population; I, irradiated population; Po1, B-lymphocytes (Lin^−^ CD45^+^ CD34^−^ CD19^+^); Po2, nuclear non-specify lineage negative cells (CD45^+^ Lin^−^ CD34^−^ CD19^−^); Po3, HSPCs (Lin^−^ CD45^+^ CD34^+^ CD19^−^); Po4, Pre-Pro B cells (Lin^−^ CD45^+^ CD34^+^CD19^+^); Po5, progenitors (Lin^−^ CD45^+^ CD45RA- CD34^+^ CD38^+^); Po6, HSCs/MPPs (Lin^−^ CD45^+^ CD45RA^−^ CD34^+^ CD38^−^); Po7, HSCs (Lin^−^ CD45^+^ CD45RA^-^ CD34^+^ CD38^−^ CD90^+^); Po8, MPPs (Lin^−^ CD45^+^ CD45RA^-^ CD34^+^ CD38^−^ CD90^−^).

**Table 3 ijms-24-06580-t003:** Summary of studies considered induction of PFGs via ionizing radiation.

Study	Type of Radiation/Dose	Cells (Cell Lines)	PFGs (Preleukemic Fusion Genes)	Results
Ito, Seyama et al., 1993 [86]	X-rays 100 Gy	HL-60	BCR-ABL1	Induction
Mizuno, Kyoizumi et al., 1997 [90]	X-rays 50 Gy	Human thyroid tissues in mice	BCR-ABL1 H4-RET (thyroid papillary carcinoma)	Induction for BCR-ABL1 (at 2 days only), induction for H4-RET
Deininger, Bose et al., 1998 [96]	γ-radiation 50, 100 Gy	HL-60 KG1	BCR-ABL1 RUNX1-RUNX1T1 DEK-NUP214	No induction for BCR-ABL and DEK-NUP214, induction for RUNX1-RUNX1T1 (only in KG1)
Spencer and Granter 1999 [87]	γ-radiation 50, 100 Gy	AML–LCL CML–LCL Non–LCL	BCR-ABL1	Induction (only in AML and CML derived LCL)
Mizuno, Iwamoto et al., 2000 [91]	X-rays 10, 50, 100 Gy	8505C Daudi G-401 HT1080	BCR-ABL1	Induction (linear dose response)
Quina, Gameiro et al., 2000 [107]	γ-radiation 10 Gy	IM9	PML-RARA	No induction
Kosik, Durdik et al., 2020 [93]	γ-radiation 0.1, 0.5, 2, 5, 10 and 30 Gy	UCB–MNCs	BCR-ABL1 ETV6-RUNX1 RUNX1-RUNX1T1 KMT2A-AFF1	No induction for all PFGs except for BCR-ABL1 at doses ≤ 0.5 Gy
Kosik, Durdik et al., 2021 [94]	γ-radiation 0.5 Gy	UCB–HSPCs	RUNX1-RUNX1T1 KMT2A-MLLT3 PML-RARA	No induction for all PFGs

## Data Availability

No new data were created or analyzed in this study. Data sharing is not applicable to this article.

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
