# Peer review of "Preleukemic Fusion Genes Induced via Ionizing Radiation"

_ijms, 2023, doi:10.3390/ijms24076580_

Round 1

Reviewer 1 Report

Preleukemic fusion genes induced by ionizing radiation by Kosik et al. (Review) ijms-2241801                          

In this work Authors review the radiologically induced appearance of several major fusion oncogenes involved in leukemia (and thyroid cancer) formation. Basic repair and mutational mechanisms, radiation dose-response relations are presented, and several major fusion oncogenes involved in leukemogenesis are discussed from a radiobiological perspective. The Paper contains mathematical calculations on risk estimation of the formation of leukemogenic translocation events as well.

Ionizing radiation as a causative agent in neoplastic transformation is well established. The study of the underlying molecular mechanisms and risk factors, and the eventual identification of people with increased risk or predisposition are important issues from a radiobiological standpoint, and also for the protection of the population in general, and thus reviewing existing work on ionizing radiation-induced oncogenic events such as oncogenic translocations/fusions can be useful.

Comments

The Reviewer thinks that the use of the term “preleukemic fusion gene” in the title and in the text is somewhat misleading, because it suggests that the topic of the paper will be closely related to preleukemia. Regarding the title, the Reviewer believes that readers will expect reading about specific characteristics of preleukemia, what differentiates preleukemia from overt leukemia in the context of those fusion oncogenes/oncoproteins and ionizing radiation etc. However, the paper actually deals with the probability of occurrence and detection of a selected number of oncogenic gene fusion events following in vitro irradiation primarily of human (already transformed) cell lines and eventually of primary human cells. Given the content of the Paper, a title like “The probability of the apparition of fusion genes induced by ionizing radiation” or something similar would be more appropriate.

In addition, although “preleukemia” is a somewhat vague notion that mostly refers to early bone marrow phases of some leukemias without overt leukemic invasion of the blood. The Reviewer is aware also of the fact that oncogenic fusion events are found at greater frequency after birth than the frequency of related leukemias, knows the paper by Nori Nakamura, and agrees with the notion that usually several oncogenic events are required for leukemic transformation, and also with the idea that single such events may lead to the benign expansion of a clonal cell population that, following additional genetic mutations, may lead to leukemia. Moreover, it is not counter-intuitive that people with such cells are at higher risk for developing haematological malignancies, either. However, the idea that such a benign clonal proliferation can be called a preleukemia needs to be substantiated. Moreover, it is also possible that the “preleukemic” fusion event itself is actually the “second hit”, preceded by another, say single nucleotide mutation in another, unidentified oncogene or tumor suppressor gene. In this sense, the “first hit” and the “second hit” may be interchangeable in terms of sequentiality.

In addition, following an extensive introduction, the paper deals mostly with in vitro irradiated cells in which the detection of a selected set of known fusion oncogenes is attempted by highly sensitive PCR-based techniques and with dosimetry-leukemogenesis considerations. There seems to be therefore a thematic gap between the title/introduction and the main body of the Paper that needs to be closed; the various parts of the paper need to be better connected, and the reader better guided among paragraphs, to facilitate reading. Dosimetry, dose-effect considerations and mathematical modelling need more explanation for the non-specialized reader as well. Maybe this could be included as Accompanying Material for the Manuscript.

It would be also very interesting if Authors could discuss the eventual appearance and detection of oncogenic mutations that in the context of the paper would be considered “secondary” compared to the fusion-generating translocations, because it is quite conceivable that such mutations (say in p53, Bcl-2, Ras, BRAF, Akt, PTEN, APC, CTNNB, cyclins, CDKs IDH, SDH etc.) can also occur in those irradiation experiments. If the first hit-second hit model is used as a conceptual basis, the discussion of this type of mutations is necessary as well. What are the elements to think that such mutations cannot precede, or be contemporaneous, with gene fusion events upon irradiation?

As also stated by the Authors, the fusion events detected following irradiation of cells/cell lines are very rare. Authors should discuss, what is the basis to construct elaborate mathematical models on the basis of events of such rarity? And to what extent is the frequency of fusions that occur in cell lines that may have very different DNA repair when compared to normal cells, pertinent for risk calculations?

Can the relative contribution/frequency of irradiation-induced oncogenic fusions estimated as compared to the same types of fusions arising by irradiation-independent mechanisms?

Lanes 5-6 : please don’t hyphenate email address (see lane 7).

Lane 8 : “The prevalence of leukemia is increasing each year.” : Could statistical data available for this be cited?

Lane 10 : “the IR induced…” : the IR-induced…

Lanes 27 and 29 : “results/resulting into” : results/resulting in

Lane 32 : “Gy” : please don’t use unexplained abbreviations in the Abstract (Gray)

Lanes 33-34 : “It is widely accepted that DSBs are the most serious lesions capable of causing cell death or cell transformation.” Please add appropriate bibliographical references here.

Lane 35 : “non-homological end joining” etc. : homologous

Lanes 35-37 : “Misrepair of the DSBs can result in formation of chromosomal aberrations, including fusion genes that in turn can lead to the development of various diseases and cancer.” : what are those various diseases, other than cancer?

Lane 59 : “Until now, several studies have supported this hypothesis” : The Reviewer is not sure whether “until” is appropriate here. If the hypothesis is still considered to be valid, “until” is probably not entirely appropriate.

Lanes 66-68 : “Interestingly, the frequency of RET-PTC1 fusions increased after exposure to IR, leading to a possibility that proximity of the two loci in interphase nucleus enhanced radiation-induced recombination between these two loci.” : Please rephrase, as it is not clear what is the additional information given in these phrase in its present form, compared to lanes 60-66. Was the frequency of RET-PTC1 fusions higher after ionizing radiation in thyroid cells, than in lymphocytes?

Lanes 69-70 : “In addition, fusion genes or rather fusion transcripts can also arise via abnormal transcription” : Please rephrase, as adequately pointed out by Authors, there are no gene fusions involved in transcript chimaeras. (Delete “fusion genes or rather”.)

Lane 73 : “Last two decades…” : In the last two decades…

Lane 75 : Please add an URL for the Mitelman database.

Lanes 80-83 : “the formation of f-circRNA made by fusion of two translocated genes is allowed by the juxtapositioning of complementary sequences in introns up- and downstream of the breakpoint region of the translocation.” : Please explain more clearly how covalent circularization and variants of circRNAs are formed. Maybe adding a Figure (eventually as Accompanying Material) would also be helpful.

Lanes 78-94 needs to be substantially improved for clarity regarding mechanism.

Lane 96 : “Leukemia is a clonal disease, which occur after genetic transformation…” : occurs

Lanes 96-97 : “genetic transformation of hematopoietic stem cell in different stages of maturation.” It is somewhat confusing to talk about hematopoietic stem cells in different stages of maturation. Is a differentiating hematopoietic stem cell still a stem cell? Please apply a more rigorous approach here for the definition of hematopoietic stem cells, and distinguish more clearly among the stem cell state, various hematopoietic progenitors and precursors.

Moreover, in the opinion of the Reviewer, it is a rather bold statement to say that leukemia is a disease of hematopoietic stem cells only, because various progenitors and precursors may also be able to undergo leukemic transformation. Please include appropriate References.

Lanes 99-101 : “ According to generally accepted hypothesis, leukemia results from at least two consecutive events. The first hit is usually prenatal chromosomal translocation that leads to preleukemic fusion gene … and the second hit usually represented by point mutations, deletions or insertions often occurs postnatally.” Please add References for this. Also, this phrase is somewhat confusing, because leukemia in adults may easily have no prenatal mutational component.

Lanes 322-323 : “CD34+ HSPCs, which are supposed to be the cells of origin for leukemia…” The idea that leukemia arises from CD34+ cells needs to be substantiated and relevant References added. And please note also, that CD34 positivity alone can hardly be considered as a strict defining criterion for hematopoietic cell stemness.

Lane 107 : “Leukemia similar to other cancers has been associated with IR.” : similarly

Lanes 109-110 : Please explain linear and quadratic dose response (and also “linear quadratic”, lane 116), as well as “curvilinear” (lane 308).

Lanes 111-112 : “can provides” : can provide

Lanes 143-150 : Regarding leukemia following therapeutic irradiation, it would be appropriate to very briefly discuss whether the frequency of leukemia was biased by shortened life expectancy due to the first (unrelated) malignancy in these cohorts, given the relatively long latency of the iatrogenic leukemia.

Lane 155 : ionizing radiation” : “ionizing radiation” (“)

Lanes 69-71 : “In addition, fusion genes or rather fusion transcripts can also arise via abnormal transcription yielding transcript chimeras, although relevant chromosomal genes are intact. These fusion transcripts can be induced by two mechanisms: (i) trans/cis-splicing [30, 71 31], and (ii) read-through [32].” Please discuss the relevance of such fusion transcripts in terms of leukemogenesis.

Lanes 99-101 : “The first hit is usually prenatal chromosomal translocation that leads to preleukemic fusion gene … and the second hit usually represented by point mutations, deletions or insertions often occurs postnatally.” : Please add appropriate References for this.

Lane 144 : “therapy related” : therapy-related

Lanes 258-262 : “A case report on three patients exposed to high level radiation from nuclear explosions during or after WW II showed that they developed almost 50 years later AML [84]. Cytogenetics revealed t(1;21), t(18;21) and t(19;21), respectively, and subsequent FISH analysis confirmed translocations with disrupted RUNX1 gene. These data indicate that disruption of RUNX1 locus may represent a key step in some leukemias induced by radiation.” This reasoning relies on the idea that these were radiation-induced leukemias. Please explain, what are the elements that allow to think that the post hoc ergo propter hoc fallacy is not at work here?

Lane 151 : “On the contrary to…” : In contrast to…?

Because of the rarity of the irradiation-induced fusion events, it would be interesting to hear about validation experiments in which fusion oncogenes are detected by the PCR techniques used for the irradiation experiments, but on non-irradiated mixtures of cell lines, in which positive and negative cells were mixed at known proportions (such as, for example, a mixture of HL-60 and K562 cells at proportions of, say 1 to 1x10-6, 1 to 1x10-7, 1 to 1x10-8 etc., for the detection of a BCR/ABL rearrangement).

Authors should discuss the estimations of the contribution of ionizing radiation (compared to all causes) to leukemogenesis and to the formation of fusion genes in greater detail.

Lane 299 : “ analysing400” : analysing 400

Lanes 167 and 243 : “HL60” : HL-60

Lanes 171-172 : “ (i) CML-specific BCR-ABL1 p210: b2a2, b3a2, (ii) atypical fusions, probably non-functional.” : (i) CML-specific BCR-ABL1 p210: b2a2, b3a2 fusions and (ii) atypical fusions, which were probably non-functional.

Lanes 172-173 : “However, this study did not mention the results of the control PCR runs.” Please specify what “control PCR runs” means here. (PCR on non-irradiated HL-60 cells?)

Lanes 174-184 : The percentage values (%) are not very informative in their present form, as it is not clear, how-many cells were used in the preparation of a PCR reaction. A given % value here represents what? How would it be possible to estimate the number of (functional) BCR/ABL-creating fusion events per, say, one million cells, or per µg cDNA, in these experiments? Idem for the various fusions in Tables 4 and 6.

Lanes 182-184 : “The data show that all LCLs contained background level of BCR-ABL1 p210. There was no BCR-ABL1 in LCL derived from the healthy individuals, while statistically significant induction of BCR-ABL1 was found in LCL derived from the AML/CML patients.” There appears to be a contradiction here: if "there was no BCR-ABL1 in LCL derived from the healthy individuals", then “1%” at 0 Gy in “non-leukemic LCL” in Table 1 is confusing.

In addition, are CML-derived LCLs guaranteed to be devoid of CML cell contamination?

The discussion of these experiments needs to be improved.

Lanes 185-186 : “human fetal tissue from thyroid” meaning human fetal thyroid tissue ?

Table 2 : The control of the experiment (0 Gy) was done at which time point (0, 2, 7 days or 2 months)?

Lanes 198-199 : “The Mizuno et al. study showed that BCR-ABL is capable of transformation not only hematopoietic cells.” Please note that these Mizuno experiments as discussed here do not show that BCR-ABL is capable of transformation not only in hematopoietic cells”. In the opinion of the Reviewer, these experiments as discussed here only show that such fusions may occur also in non-hematopoietic cells. Is transformation (i.e.: enhanced clonal growth) demonstrated here?

Lane 202 : “irradiated to X-rays” : irradiated with X-rays (or: “exposed to X-rays”?)

Lane 210 : “They found that all CML patients were Ph+ contained BCR-ABL1 p210 fusion protein.” : Please rephrase. (…patients who were Ph+ contained…?)

Lane 214 : “and subpopulation of HSPCs” : which subpopulation?

Lanes 220-221 : “…a low but statistically significant inducibility of BCR-ABL p190 at low doses up to 50 cGy in comparison with high doses above 50 cGy.” It is not clear, what “in comparison” means here: was induction more marked at lower (i.e.: <50 cGy) radiation doses?

Lane 223 : “HSPC” : ? Stem- and progenitor cells?

Table 5 : Please discuss, what the p value (*= p < 0.0001) means here (as well as the absence thereof for the other values).

Lanes 256-257 : “This data suggest that IR is capable of inducing PFGs with different incidence in cell populations of different susceptibility.” What is the relevance of measuring the occurrence of post-irradiation fusion events in cell lines that are already leukemic? It is quite conceivable that the efficacy of various repair mechanisms in such cells are rather different when compared to non-transformed cells.

Lane 296 : “in lymphoid-cell line” : in a lymphoid-cell line

Lanes 302-303 and Table 6 : “In summary, similar incidence of PML-RARA was observed between irradiated and control samples. Thus, the data indicate that the PML and RARA genes are not particularly susceptible to the clastogenic effect of IR. How about the 0.75% value for M2-R5/R8 at 10 Gy?

Can it be stated with certainty that APL arises from the transformation of a bona fide HSC, rather than of an early myeloid progenitor?

Lanes 324-325 : “The factor 4 accounts for the fact that the cell has diploid number of chromosomes” : please explain.

Lane 348 : “Using the estimated by us number of 1.18x10-4” : Using the number of 1.18x10-4 estimated by us…

Lanes 305-373 : The mathematical estimations shown here are not presented with sufficient clarity, are not immediately convincing and appear rather speculative. This part needs major improvement before the Manuscript can be considered for publication; Supplementary Material could be used for enhanced discussion and mathematics. Alternatively, submission to a specialized radiobiology journal may be considered.

Author Response

The authors are cordially thankful to this reviewer for the comments, which helped to improve the manuscript and were addressed in both this response (R:) and revised text.

Preleukemic fusion genes induced by ionizing radiation by Kosik et al. (Review) ijms-2241801                          

In this work Authors review the radiologically induced appearance of several major fusion oncogenes involved in leukemia (and thyroid cancer) formation. Basic repair and mutational mechanisms, radiation dose-response relations are presented, and several major fusion oncogenes involved in leukemogenesis are discussed from a radiobiological perspective. The Paper contains mathematical calculations on risk estimation of the formation of leukemogenic translocation events as well.

Ionizing radiation as a causative agent in neoplastic transformation is well established. The study of the underlying molecular mechanisms and risk factors, and the eventual identification of people with increased risk or predisposition are important issues from a radiobiological standpoint, and also for the protection of the population in general, and thus reviewing existing work on ionizing radiation-induced oncogenic events such as oncogenic translocations/fusions can be useful.

Comments

The Reviewer thinks that the use of the term “preleukemic fusion gene” in the title and in the text is somewhat misleading, because it suggests that the topic of the paper will be closely related to preleukemia. Regarding the title, the Reviewer believes that readers will expect reading about specific characteristics of preleukemia, what differentiates preleukemia from overt leukemia in the context of those fusion oncogenes/oncoproteins and ionizing radiation etc. However, the paper actually deals with the probability of occurrence and detection of a selected number of oncogenic gene fusion events following in vitro irradiation primarily of human (already transformed) cell lines and eventually of primary human cells. Given the content of the Paper, a title like “The probability of the apparition of fusion genes induced by ionizing radiation” or something similar would be more appropriate.

In addition, although “preleukemia” is a somewhat vague notion that mostly refers to early bone marrow phases of some leukemias without overt leukemic invasion of the blood. The Reviewer is aware also of the fact that oncogenic fusion events are found at greater frequency after birth than the frequency of related leukemias, knows the paper by Nori Nakamura, and agrees with the notion that usually several oncogenic events are required for leukemic transformation, and also with the idea that single such events may lead to the benign expansion of a clonal cell population that, following additional genetic mutations, may lead to leukemia. Moreover, it is not counter-intuitive that people with such cells are at higher risk for developing haematological malignancies, either. However, the idea that such a benign clonal proliferation can be called a preleukemia needs to be substantiated.

Moreover, it is also possible that the “preleukemic” fusion event itself is actually the “second hit”, preceded by another, say single nucleotide mutation in another, unidentified oncogene or tumor suppressor gene. In this sense, the “first hit” and the “second hit” may be interchangeable in terms of sequentiality.

R2: We agree with this comment and revised the text accordingly

In addition, following an extensive introduction, the paper deals mostly with in vitro irradiated cells in which the detection of a selected set of known fusion oncogenes is attempted by highly sensitive PCR-based techniques and with dosimetry-leukemogenesis considerations. There seems to be therefore a thematic gap between the title/introduction and the main body of the Paper that needs to be closed; the various parts of the paper need to be better connected, and the reader better guided among paragraphs, to facilitate reading.

R3: Unfortunately, only few in vivo studies are available for radiation-induced PFG. In response to this comment, we have retrieved a new studies, which have analysed aberrant clones and hot spots in radiation induced chromosomal aberrations in vivo. mFISH has been used providing a possibility to screen multiple chromosomal aberrations and detecting aberrant clones and hot spots. These studies indicated possible correlation with IR-induced aberrations and PFGs. The text has been revised accordingly.

Dosimetry, dose-effect considerations and mathematical modelling need more explanation for the non-specialized reader as well. Maybe this could be included as Accompanying Material for the Manuscript.

It would be also very interesting if Authors could discuss the eventual appearance and detection of oncogenic mutations that in the context of the paper would be considered “secondary” compared to the fusion-generating translocations, because it is quite conceivable that such mutations (say in p53, Bcl-2, Ras, BRAF, Akt, PTEN, APC, CTNNB, cyclins, CDKs IDH, SDH etc.) can also occur in those irradiation experiments. If the first hit-second hit model is used as a conceptual basis, the discussion of this type of mutations is necessary as well. What are the elements to think that such mutations cannot precede, or be contemporaneous, with gene fusion events upon irradiation?

R5: Epidemiological studies have suggested that leukemogenesis usually occurs as a multistep process with the initiating event arising in utero and subsequent genetic “hits” (mutations) occur postnatally (Greeves, 1993; Sandler and Ross, 1997; Greeves, 1997). This concept was confirmed for childhood leukemia studies on pairs of monozygotic monochorionic twins which shared identical breakpoints in introns of genes involved in chromosomal translocation (Greeves, 1999). As mentioned previously, leukaemia is a multistep process and additional mutations that may clearly contribute to disease evolution, progression and malignancy are always present. However, no other mutations were studied in the retrieved by us studies on radiation-induced PFG and thus we consider such discussion as a premature one.

As also stated by the Authors, the fusion events detected following irradiation of cells/cell lines are very rare. Authors should discuss, what is the basis to construct elaborate mathematical models on the basis of events of such rarity? And to what extent is the frequency of fusions that occur in cell lines that may have very different DNA repair when compared to normal cells, pertinent for risk calculations?

R6: Thank you for this comment, we have done our best to better describe a mathematical model and are confident that it is well understandable in the revised text.

Can the relative contribution/frequency of irradiation-induced oncogenic fusions estimated as compared to the same types of fusions arising by irradiation-independent mechanisms?

R7: Thank you, it is very interesting question, which should be considered in a special review but we assume it is out of the scope of the current manuscript, which is focused on radiation-induced PFG.

Lanes 5-6 : please don’t hyphenate email address (see lane 7).

R8: E-mail address has been provided according the journal’s rules.

Lane 8 : “The prevalence of leukemia is increasing each year.” : Could statistical data available for this be cited?

R9: The small trend of growing incidence is visible. Here, we attached the reference with link: Surveillance, Epidemiology, and End Results (SEER) 8 registries, National Cancer Institute, 2022. https://cancerstatisticscenter.cancer.org/cancer-site/Leukemia/nWHbpgnv

However, no references are usually provided in abstracts. The text has been edited accordingly.

Lane 10 : “the IR induced…” : the IR-induced…

R10: The text was modified according to the reviewer's suggestion.

Lanes 27 and 29 : “results/resulting into” : results/resulting in

R11: The text was modified according to the reviewer's suggestion.

Lane 32 : “Gy” : please don’t use unexplained abbreviations in the Abstract (Gray)

R12: The abbreviation Gy was not present in the abstract. The abbreviation (Gy) was explained in the first place in the manuscript, in the “Ionizing radiation and DNA damage” section. All abbreviations were explained in the abstract. The text was revised accordingly.

Lanes 33-34 : “It is widely accepted that DSBs are the most serious lesions capable of causing cell death or cell transformation.” Please add appropriate bibliographical references here.

R13: The references were added to the manuscript (Cannan et al 2016), (Thadathil et al 2019).

Lane 35 : “non-homological end joining” etc. : homologous

R14: The text was revised accordingly.

Lanes 35-37 : “Misrepair of the DSBs can result in formation of chromosomal aberrations, including fusion genes that in turn can lead to the development of various diseases and cancer.” : what are those various diseases, other than cancer?

R15: Family-inherited fusion genes have been known to be associated with human diseases for decades, however, only a small number of them have been discovered so far. As an example, a PTK2-THOC2 gene fusion (involving chromosomes Xq25 and 8q24) with THOC2 expression knockdown has been described in a pediatric patient with psychomotor retardation and congenital cerebellar hypoplasia (DiGregorio et al, 2013). The presence of hereditary fusion genes (HFG) supported Zhuo (2022) who in 37 pairs of monozygotic twins (by RNAseq) identified 660 inter-chromosomal HFG of total 1,180 HGF.

Lane 59 : “Until now, several studies have supported this hypothesis” : The Reviewer is not sure whether “until” is appropriate here. If the hypothesis is still considered to be valid, “until” is probably not entirely appropriate.

R16: Text was modified accordingly.

Lanes 66-68 : “Interestingly, the frequency of RET-PTC1 fusions increased after exposure to IR, leading to a possibility that proximity of the two loci in interphase nucleus enhanced radiation-induced recombination between these two loci.” : Please rephrase, as it is not clear what is the additional information given in these phrase in its present form, compared to lanes 60-66. Was the frequency of RET-PTC1 fusions higher after ionizing radiation in thyroid cells, than in lymphocytes?

R17: The information in this sentence was very similar as compared to the lanes 60-66. Therefore, this sentence was deleted while moving the references to the 66 lane.

The study of Nikiforova et al. used two-colour fluorescence in situ hybridization and three-dimensional microscopy to map the positions of the RET and H4 loci within interphase nuclei. At least one pair of RET and H4 was juxtaposed in 35% of normal human thyroid cells and in 21% of peripheral blood lymphocytes. The frequency of RET-PTC fusions was not studied in irradiated lymphocytes.

Lanes 69-70 : “In addition, fusion genes or rather fusion transcripts can also arise via abnormal transcription” : Please rephrase, as adequately pointed out by Authors, there are no gene fusions involved in transcript chimaeras. (Delete “fusion genes or rather”.)

R18: The text was rephrased accordingly. “Fusion genes or rather” was deleted from the sentence.

Lane 73 : “Last two decades…” : In the last two decades…

R19: The text was modified according to the reviewer's suggestion.

Lane 75 : Please add an URL for the Mitelman database.

R20: URL was added to the manuscript.

Lanes 80-83 : “the formation of f-circRNA made by fusion of two translocated genes is allowed by the juxtapositioning of complementary sequences in introns up- and downstream of the breakpoint region of the translocation.” : Please explain more clearly how covalent circularization and variants of circRNAs are formed. Maybe adding a Figure (eventually as Accompanying Material) would also be helpful.

Lanes 78-94 needs to be substantially improved for clarity regarding mechanism.

R21: The text has been revised accordingly. Thank you for your suggestions, but we assume that mechanisms how covalent circularization and variants of circRNAs are formed are out of the scope of the current manuscript, which is focused on radiation-induced PFG.

Lane 96 : “Leukemia is a clonal disease, which occur after genetic transformation…” : occurs

R22: Text was modified accordingly.

Lanes 96-97 : “genetic transformation of hematopoietic stem cell in different stages of maturation.” It is somewhat confusing to talk about hematopoietic stem cells in different stages of maturation. Is a differentiating hematopoietic stem cell still a stem cell? Please apply a more rigorous approach here for the definition of hematopoietic stem cells, and distinguish more clearly among the stem cell state, various hematopoietic progenitors and precursors.

Moreover, in the opinion of the Reviewer, it is a rather bold statement to say that leukemia is a disease of hematopoietic stem cells only, because various progenitors and precursors may also be able to undergo leukemic transformation. Please include appropriate References.

R23: Text was modified according to the reviewer's suggestion. The meaning of this sentence was better described. The phrase of “hematopoietic stem cell in different stages of maturation” was replaced by “hematopoietic stem, progenitor and precursor cells”. References were added in the manuscript (Anderson et al. 2011, Castor et al. 2005, Long et al. 2022).

Lanes 99-101 : “ According to generally accepted hypothesis, leukemia results from at least two consecutive events. The first hit is usually prenatal chromosomal translocation that leads to preleukemic fusion gene … and the second hit usually represented by point mutations, deletions or insertions often occurs postnatally.” Please add References for this. Also, this phrase is somewhat confusing, because leukemia in adults may easily have no prenatal mutational component.

R24: The references were added to the manuscript (Greaves 2003, Wiemels 2008). Few sentences with the references were added to this chapter to indicate the difference between child and adult leukaemia (Marcotte 2021, Roberts 2018).

Lanes 322-323 : “CD34+ HSPCs, which are supposed to be the cells of origin for leukemia…” The idea that leukemia arises from CD34+ cells needs to be substantiated and relevant References added. And please note also, that CD34 positivity alone can hardly be considered as a strict defining criterion for hematopoietic cell stemness.

R25: ALL and AML leukaemia are very heterogeneous diseases. CD34 marker is often used as a marker of not only HSC but also progenitor cells. Definitely CD34 marker alone cannot be a marker for cell stemness, but several studies demonstrated that leukemia-initiating cells with self-renewal capacity in human ALL or AML resided in CD34+ population. (Lapidot et al., 1994, Bonnet and Dick 1997, A. Blair et al 1998, Kong et al. 2008, A van Rhenen 2007).

Lane 107 : “Leukemia similar to other cancers has been associated with IR.” : similarly

R26: Text was modified according to the reviewer's suggestion.

Lanes 109-110 : Please explain linear and quadratic dose response (and also “linear quadratic”, lane 116), as well as “curvilinear” (lane 308).

R27: The linear and linear quadratic models are commonly accepted models in radiation biology. This linear quadratic model is based on the hypothesis of dual radiation action. The linear component is produced by a single act of radiation and the quadratic component is the result of two independent acts of ionising radiation. In a range of very low doses the contribution of quadratic components is negligible, so the dose response can be approximated by linear components. For higher doses quadratic component is increased. Curvilinear was replaced with linear quadratic.

Lanes 111-112 : “can provides” : can provide

R28: Text was modified according to the reviewer's suggestion.

Lanes 143-150 : Regarding leukemia following therapeutic irradiation, it would be appropriate to very briefly discuss whether the frequency of leukemia was biased by shortened life expectancy due to the first (unrelated) malignancy in these cohorts, given the relatively long latency of the iatrogenic leukemia.

R29: The text was revised accordingly.

Lane 155 : ionizing radiation” : “ionizing radiation” (“)

R30: Text was modified according to the reviewer's suggestion.

Lanes 69-71 : “In addition, fusion genes or rather fusion transcripts can also arise via abnormal transcription yielding transcript chimeras, although relevant chromosomal genes are intact. These fusion transcripts can be induced by two mechanisms: (i) trans/cis-splicing [30, 71 31], and (ii) read-through [32].” Please discuss the relevance of such fusion transcripts in terms of leukemogenesis.

R31: The relevance of fusion transcripts in terms of leukemogenesis was described.

Lanes 99-101 : “The first hit is usually prenatal chromosomal translocation that leads to preleukemic fusion gene … and the second hit usually represented by point mutations, deletions or insertions often occurs postnatally.” : Please add appropriate References for this.

R32: The references were added to the manuscript (Greaves 2003, Wiemels 2008).

Lane 144 : “therapy related” : therapy-related

R33: The text was modified accordingly.

 Lanes 258-262 : “A case report on three patients exposed to high level radiation from nuclear explosions during or after WW II showed that they developed almost 50 years later AML [84]. Cytogenetics revealed t(1;21), t(18;21) and t(19;21), respectively, and subsequent FISH analysis confirmed translocations with disrupted RUNX1 gene. These data indicate that disruption of RUNX1 locus may represent a key step in some leukemias induced by radiation.” This reasoning relies on the idea that these were radiation-induced leukemias. Please explain, what are the elements that allow to think that the post hoc ergo propter hoc fallacy is not at work here?

R34: We agree with the opinion of reviewer and this part was deleted from the manuscript.

Lane 151 : “On the contrary to…” : In contrast to…?

R35: Text was modified according to the reviewer's suggestion.

Because of the rarity of the irradiation-induced fusion events, it would be interesting to hear about validation experiments in which fusion oncogenes are detected by the PCR techniques used for the irradiation experiments, but on non-irradiated mixtures of cell lines, in which positive and negative cells were mixed at known proportions (such as, for example, a mixture of HL-60 and K562 cells at proportions of, say 1 to 1x10-6, 1 to 1x10-7, 1 to 1x10-8 etc., for the detection of a BCR/ABL rearrangement).

R36: Thank you very much for this extremely interesting suggestion, which warrants to be explored in a separate study.

Authors should discuss the estimations of the contribution of ionizing radiation (compared to all causes) to leukemogenesis and to the formation of fusion genes in greater detail.

R37: The text was revised accordingly to discuss the estimations of the contribution of ionizing radiation in greater detail.

Lane 299 : “ analysing400” : analysing 400

R38: The text was modified according to the reviewer's suggestion.

Lanes 167 and 243 : “HL60” : HL-60

R39: The text was revised accordingly.

Lanes 171-172 : “ (i) CML-specific BCR-ABL1 p210: b2a2, b3a2, (ii) atypical fusions, probably non-functional.” : (i) CML-specific BCR-ABL1 p210: b2a2, b3a2 fusions and (ii) atypical fusions, which were probably non-functional.

R40: Text was modified according to the reviewer's suggestion.

Lanes 172-173 : “However, this study did not mention the results of the control PCR runs.” Please specify what “control PCR runs” means here. (PCR on non-irradiated HL-60 cells?)

R41: In the study of Ito et al., negativity of non-irradiated HL-60 cells has been confirmed only by a single PCR run (in one reaction). In contrast, in case of irradiated HL-60 total RNA from 108 cells was analysed by 41 independent RT-PCRs.

Lanes 174-184 : The percentage values (%) are not very informative in their present form, as it is not clear, how-many cells were used in the preparation of a PCR reaction. A given % value here represents what? How would it be possible to estimate the number of (functional) BCR/ABL-creating fusion events per, say, one million cells, or per µg cDNA, in these experiments? Idem for the various fusions in Tables 4 and 6.

 R42: In the tables we show the % of IR-induced PFGs positive reactions from the total number of positive reactions. The number of reactions, cells, µg cDNA is in detail described in individual publications.

Lanes 182-184 : “The data show that all LCLs contained background level of BCR-ABL1 p210. There was no BCR-ABL1 in LCL derived from the healthy individuals, while statistically significant induction of BCR-ABL1 was found in LCL derived from the AML/CML patients.” There appears to be a contradiction here: if "there was no BCR-ABL1 in LCL derived from the healthy individuals", then “1%” at 0 Gy in “non-leukemic LCL” in Table 1 is confusing.

In addition, are CML-derived LCLs guaranteed to be devoid of CML cell contamination?

The discussion of these experiments needs to be improved.

R43: We have corrected this part of the manuscript. This study showed the background level of BCR-ABL1 (unirradiated, 0 Gy). Referring to other studies, this study assumes that the detection of residual BCR-ABL positivity within CML-derived LCL was unlikely and did not produce false-positive PCR results. However contamination still cannot be excluded.

Lanes 185-186 : “human fetal tissue from thyroid” meaning human fetal thyroid tissue ?

R44: The text was revised accordingly.

Table 2 : The control of the experiment (0 Gy) was done at which time point (0, 2, 7 days or 2 months)?

R45: The Mizuno study did not mention exact time point for detection of control stating “we could not find any positive bands of BCR-ABL and H4-RET genes in non-irradiated human thyroid tissues”. Therefore, we added “before IR” to the Table 2.

Lanes 198-199 : “The Mizuno et al. study showed that BCR-ABL is capable of transformation not only hematopoietic cells.” Please note that these Mizuno experiments as discussed here do not show that BCR-ABL is capable of transformation not only in hematopoietic cells”. In the opinion of the Reviewer, these experiments as discussed here only show that such fusions may occur also in non-hematopoietic cells. Is transformation (i.e.: enhanced clonal growth) demonstrated here?

R46: The opinion of reviewer is correct – Mizuno´s study showed that BCR-ABL fusions may occur also in non-hematopoietic cells; the transformation of those cells was not demonstrated. The discussion of this study was revised.

Lane 202 : “irradiated to X-rays” : irradiated with X-rays (or: “exposed to X-rays”?)

R47: The text was modified according to the reviewer's suggestion.

Lane 210 : “They found that all CML patients were Phcontained BCR-ABL1 p210 fusion protein.” : Please rephrase. (…patients who were Ph+ contained…?)

R48: The text was rephrased accordingly.

Lane 214 : “and subpopulation of HSPCs” : which subpopulation?

R59: The text was corrected and subpopulations of HSPCs with specific CD markers were explained under the Table 4.

Lanes 220-221 : “…a low but statistically significant inducibility of BCR-ABL p190 at low doses up to 50 cGy in comparison with high doses above 50 cGy.” It is not clear, what “in comparison” means here: was induction more marked at lower (i.e.: <50 cGy) radiation doses?

R50: Yes, the induction of BCR-ABL1 was more marked at lower radiation doses (i.e.: <50 cGy). As far as low and high doses may induce different response, we also analyzed the low doses (≤ 50 cGy) and the high doses (≥ 200 cGy) separately. The amount of BCR-ABL positive samples increased from 2 in controls to 6 in samples irradiated with low doses (ANOVA, p = 0.021). The other PFGs showed statistically significant induction for neither low nor high doses.

Lane 223 : “HSPC” : ? Stem- and progenitor cells?

R51: The abbreviation HSPCs was explained at the first occasion in the chapter “PFGs induced by IR”.

Table 5 : Please discuss, what the p value (*= p < 0.0001) means here (as well as the absence thereof for the other values).

R52: This p value was the results of chi-squared test. Statistically significant effect was observed only for RUNX1-RUNX1T1 in KG1 cell line and not for other cell lines and doses.

Lanes 256-257 : “This data suggest that IR is capable of inducing PFGs with different incidence in cell populations of different susceptibility.” What is the relevance of measuring the occurrence of post-irradiation fusion events in cell lines that are already leukemic? It is quite conceivable that the efficacy of various repair mechanisms in such cells are rather different when compared to non-transformed cells.

R53: We agree with the reviewer that these cells could have different DNA repair mechanisms. Karyotypic analysis of both HL-60 and KG1 cell lines used in this study confirmed the absence of studied translocations.

HL-60 cells are promyeoloblasts isolated from the peripheral blood by leukopheresis from a 36-year-old white female with acute promyelocytic leukemia Þ the cell type: promyeloblasts.

The KG-1 cell line is made up of macrophages isolated from a bone marrow aspirate obtained from a 59-year-old white male with erythroleukemia that evolved into acute myelogenous leukemia. KG-1 is used in cancer, immunology, and toxicology research Þ the cell type: macrophages.

Indeed, the efficacy of various repair pathways is different in leukemic cell lines as compared to non-transformed cells and also within each of these two categories. Deiniger´s study just showed the significant difference in induction of specific fusion genes by high-dose IR in different cell types (promyeloblasts vs macrophages) originated from different types of leukemia (promyelotic leukemia vs erythroleukemia).

Lane 296 : “in lymphoid-cell line” : in a lymphoid-cell line

R54: The text was modified according to the reviewer's suggestion.

Lanes 302-303 and Table 6 : “In summary, similar incidence of PML-RARA was observed between irradiated and control samples. Thus, the data indicate that the PML and RARA genes are not particularly susceptible to the clastogenic effect of IR. How about the 0.75% value for M2-R5/R8 at 10 Gy?

R55: In this study, the p-values for comparison of PML-RARA between irradiated and control samples were higher than 0.05 (the p-value was 0.616 for M2-R5/R8 at 10 Gy). It means that no differences were observed between irradiated and control samples. P-values were added to the manuscript.

Can it be stated with certainty that APL arises from the transformation of a bona fide HSC, rather than of an early myeloid progenitor?

R56: APL typically displays most of the following immunophenotypic features, including high side scatter; positivity for CD13, CD33, and CD117; and absent expression of CD34, HLA-DR, CD10, CD11a, CD11b, CD11c, and CD18 (Horna et al. 2014). However, it is well documented that CD2+, CD56+, and CD34+ APL immunophenotypes are associated with lower overall survival (OS) rate, shorter remission, decreased incidence of remission, and increased incidence of early death, respectively (Ahmed et al. 2012, Albano et al. 2006).

Lanes 324-325 : “The factor 4 accounts for the fact that the cell has diploid number of chromosomes” : please explain.

R57: Number of base pairs in genome (haploid) must be multiplied by 2 because of diploid number of chromosomes and further by 2 because of 2 sister chromatids in each chromosomes in mitosis.

Lane 348 : “Using the estimated by us number of 1.18x10-4…” : Using the number of 1.18x10-4 estimated by us…

R58: The text was modified according to the reviewer's suggestion.

Lanes 305-373 : The mathematical estimations shown here are not presented with sufficient clarity, are not immediately convincing and appear rather speculative. This part needs major improvement before the Manuscript can be considered for publication; Supplementary Material could be used for enhanced discussion and mathematics. Alternatively, submission to a specialized radiobiology journal may be considered.

R59: The text was revised accordingly and we are confident that mathematical estimations shown here are now presented with sufficient clarity.

Submission Date

09 February 2023

Date of this review

19 Feb 2023 20:53:33

Reviewer 2 Report

In the present manuscript, the authors comprehensively summarize the findings on radiation-induced preleukemic fusion genes (PFGs). This review article will be helpful for the readers to understand the mechanisms.  Just a few minor comments that would improve an already very nice manuscript.

1. (Table 7) Several studies observed the induction of PFGs using the leukemic cell lines, while no studies did not see the effect using normal cells. I think this point is worth discussing somewhere in the manuscript.

2. (Table 7) Please consider describing the word of "PFGs" and "Cells" more in the table since it is difficult to understand the words if the readers just only see it. 

Author Response

The authors are cordially thankful to this reviewer for the comments, which helped to improve the manuscript and were addressed in both this response (R:) and revised text.

In the present manuscript, the authors comprehensively summarize the findings on radiation-induced preleukemic fusion genes (PFGs). This review article will be helpful for the readers to understand the mechanisms.  Just a few minor comments that would improve an already very nice manuscript.

  1. (Table 7) Several studies observed the induction of PFGs using the leukemic cell lines, while no studies did not see the effect using normal cells. I think this point is worth discussing somewhere in the manuscript.

R1: As a matter of fact, only leukemic cell lines were studied by other research groups in respect to induction of PFG by ionizing radiation. Only two studies, both by our research group, have analyzed response of normal cells, one showing response at low doses while another one not detecting any effect. Thus, we assume that more studies with normal cells are warranted to discuss a possible difference.

  1. (Table 7) Please consider describing the word of "PFGs" and "Cells" more in the table since it is difficult to understand the words if the readers just only see it. 

R2: The words "PFGs" and "Cells" have been better described in the parenthesis (Table 7).

Submission Date

09 February 2023

Date of this review

18 Feb 2023 09:44:19

Round 2

Reviewer 1 Report

Comments to the revised version

Lane 35 : “non-homological” ?

Lanes 78-79 : “FTs encoded by chromosomal translocation t(2;13) are a driving force of alveolar rhabdomyosarcoma, while the identical fusion chimeras are found in muscle biopsies in aborted fetuses, i.e. they have developmental role in normal cells.” : Please explain and add References.

Lanes 134-135 : Please explain what T, C, alpha, beta and D denote in the equation.

Lane 140 : “for the most solid cancers” : for most solid cancers

Lanes 174-176: What does 33.6% mean here? And please note that 144 is the 0.215% of 66896.

Lane 178: “increased to 72%” or “increased by 72%”?

Lane 194 : Regarding to?

Lane 250: “non-stimulated” : to divide?

Lane 309: involving the MLL gene…

The Reviewer finds that the mathematical modelling presented in the Manuscript is still not explained satisfactorily for non-specialist readers. For example: “Finally, we compared the estimated number of individuals with the number of affected individuals with CML and general population” is not clear. Lanes 350-351: “Under the assumptions that the probability for a chromosome to participate in a radiation-induced chromosome exchange aberration is proportional to its DNA content and random distribution of IR-induced break points” This may be valid for any random translocation. It is not clear, however, how the length of a chromosome influences the probability of a specific oncogenic translocation. Do the Authors imply, for example, that if the 9th and/or the 22th chromosomes were two times longer, CML were two (or four) times less frequent in the population (after equal irradiation)? If length is important, isn’t it the length of the breakpoint cluster region that should matter?

Lane 458: “but by secondary mutations…”

Lane 464: “accidentally”

Lane 466: “repeatedly many years”: ?

Lane 468: “victims”

Lane 470: “cells. Namely…” : cells, namely…

Lane 479: “was found”?

Lanes 489-490: “This seemingly confusing result is likely due to a very low detected copy number PFGs (1 x 10-5), which is close to the sensitivity threshold of the R-T qPCR”. In light of this sentence please discuss, what is the validity of the PCR results and of the quantitative deductions based on them and presented in this work in general?

Lane 507: “very low doses 0.1 Gy” Can 0.1 Gy be considered a very low dose of ionizing radiation?

Author Response

Again, the authors are very thankful to this Reviewer, whose comments have helped to improve the manuscript.

Comments to the revised version

Lane 35 : “non-homological” ?

R1: The text was modified accordingly: “non-homologous end joining”.

Lanes 78-79 : “FTs encoded by chromosomal translocation t(2;13) are a driving force of alveolar rhabdomyosarcoma, while the identical fusion chimeras are found in muscle biopsies in aborted fetuses, i.e. they have developmental role in normal cells.” : Please explain and add References.

R2: PAX3-FOXO1 and JAZF1-SUZ12 represent the first two cases where identical fusion products are found in both normal physiology and cancer malignancies. It is intriguing that identical fusion RNAs are generated by trans-splicing in normal cells as those produced by chromosomal rearrangement in neoplasia cells. The authors hypothesised that hyperactive or inappropriate trans-splicing of RNA may contribute to neoplastic transformation of cells, while in normal cells, the trans-splicing is regulated and the resultant chimeric RNAs have some physiological functions during different stages of development. References were added. (Yuan H. et al. 2013, Li, H. et al 2009)

Lanes 134-135 : Please explain what T, C, alpha, beta and D denote in the equation.

R3: Y - is the yield, D is the dose, C is the control (background frequency), α is the linear coefficient, and β is the dose squared coefficient. The text was modified accordingly.

Lane 140 : “for the most solid cancers” : for most solid cancers

R4: The text was modified according to the reviewer’s suggestion.

Lanes 174-176: What does 33.6% mean here? And please note that 144 is the 0.215% of 66896.

R5: In the study of Wright et al. a total of 199,268 individuals, including 66,896 (33,6%) who received pelvic radiotherapy and 132,372 (66.4%) not treated with radiation, were identified. 0.0022% was corrected to 0.215%. The text was modified accordingly.

Lane 178: “increased to 72%” or “increased by 72%”?

R6: The text was modified accordingly: “by 72%”

Lane 194 : Regarding to?

R7: The text was revised accordingly: “According to”

Lane 250: “non-stimulated” : to divide?

R8: Yes, the cells were not stimulated to cell division.

Lane 309: involving the MLL gene…

R9: The text was modified accordingly.

The Reviewer finds that the mathematical modelling presented in the Manuscript is still not explained satisfactorily for non-specialist readers. For example: “Finally, we compared the estimated number of individuals with the number of affected individuals with CML and general population” is not clear.

Lanes 350-351: “Under the assumptions that the probability for a chromosome to participate in a radiation-induced chromosome exchange aberration is proportional to its DNA content and random distribution of IR-induced break points” This may be valid for any random translocation. It is not clear, however, how the length of a chromosome influences the probability of a specific oncogenic translocation. Do the Authors imply, for example, that if the 9th and/or the 22th chromosomes were two times longer, CML were two (or four) times less frequent in the population (after equal irradiation)? If length is important, isn’t it the length of the breakpoint cluster region that should matter?

R10: The number of individuals was estimated mathematically to be equal 0.92 or about one individual. We have revised the text accordingly.

Under the assumption of equal random distribution of IR-induced DNA strand breaks, the incidence of PFGs is proportional to the length of the PFG break point cluster and reversely proportional to the genome length. We apologise for wording in the original text, which has resulted in confusion. The revised text was modified accordingly.

Lane 458: “but by secondary mutations…”

R11: Text was modified accordingly.

Lane 464: “accidentally”

R12: Text was modified accordingly.

Lane 466: “repeatedly many years”: ?

R13: Text was modified accordingly: “repeatedly for many years” The cells were analysed in 2016-2020 years.

Lane 468: “victims”

R14: victim – Lymphocytes only from one patient were analysed.

Lane 470: “cells. Namely…” : cells, namely…

R15: The text was modified according the reviewer suggestion.

Lane 479: “was found”?

R16: It was changed to “was detected”

Lanes 489-490: “This seemingly confusing result is likely due to a very low detected copy number PFGs (1 x 10-5), which is close to the sensitivity threshold of the R-T qPCR”. In light of this sentence please discuss, what is the validity of the PCR results and of the quantitative deductions based on them and presented in this work in general?

R17: In these studies (Table 7) IR-induced PFGs have been analysed by either nested RT PCR or RT qPCR and the PCR products were used as templates for sequencing. Sensitivity of nested RT PCR and RT qPCR has been reported in the range of 10-6 to 10-7 and 10-4 to 10-5, respectively. In most studies (Table 7) RNA was amplified separately in individual tubes containing 106-7 cells. Thus the frequencies of induced fusion genes were given as the number of tubes that showed PCR amplification per total number of tubes or reactions. Because only few tubes from the total number of tested tubes usually contained PFGs, it can be assumed that the frequency of IR-induced PFGs is very low and probably on the border of PCR sensitivity. Mizuno et al suggested that the induction frequency for BCR-ABL1 was in one study 1.1 x 10-7 in human thyroid cells after 50Gy and in another study 1x10-9, 1.4x10-8 and 4.2x10-8 at 10, 50 and 100Gy, respectively in four different cell lines: 8505C, Daudi, G401, and HT1080. The text was revised accordingly.

Lane 507: “very low doses ≤ 0.1 Gy” Can 0.1 Gy be considered a very low dose of ionizing radiation?

R18: The word “very” was deleted. According the study of (Little et al. 2022) conventional definitions of low doses is <0.1 Gy, and moderate doses 0.1–1 Gy.

Submission Date

09 February 2023

Date of this review

20 Mar 2023 15:53:30
